# Cavity-enhanced microwave readout of a solid-state spin sensor

Erik R. Eisenach [1,2], John F. Barry[2✉], Michael F. O'Keeffe[2], Jennifer M. Schloss[2], Matthew H. Steinecker[2], Dirk R. Englund [1] & Danielle A. Braje[2]

Overcoming poor readout is an increasingly urgent challenge for devices based on solid-state spin defects, particularly given their rapid adoption in quantum sensing, quantum information, and tests of fundamental physics. However, in spite of experimental progress in specific systems, solid-state spin sensors still lack a universal, high-fidelity readout technique. Here we demonstrate high-fidelity, room-temperature readout of an ensemble of nitrogen-vacancy centers via strong coupling to a dielectric microwave cavity, building on similar techniques commonly applied in cryogenic circuit cavity quantum electrodynamics. This strong collective interaction allows the spin ensemble's microwave transition to be probed directly, thereby overcoming the optical photon shot noise limitations of conventional fluorescence readout. Applying this technique to magnetometry, we show magnetic sensitivity approaching the Johnson–Nyquist noise limit of the system. Our results pave a clear path to achieve unity readout fidelity of solid-state spin sensors through increased ensemble size, reduced spin-resonance linewidth, or improved cavity quality factor.

[1] Massachusetts Institute of Technology, Cambridge, MA, USA. [2] MIT Lincoln Laboratory, Lexington, MA, USA. ✉email: john.barry@ll.mit.edu

Quantum devices employing optically active solid-state spin ensembles promise broad utility[1–5] but are plagued by poor readout[6]. Conventional spin readout via optical excitation and fluorescence detection destroys the information stored by a spin defect with only a few scattered photons. Imperfect optical collection then ensures that on average less than one fluorescence photon is typically detected per spin[1]. Moreover, spin fluorescence contrast (i.e., the normalized difference in signal between spin states) is far below unity, which further reduces the quantum information that conventional readout can extract from a given spin. Hence, quantum sensors employing solid-state spin ensembles with conventional optical readout exhibit sensitivities much worse than the spin-projection noise limit, with readout fidelities $\mathcal{F} \ll 1$ limited by shot noise on the detected fluorescence[6]. Here $\mathcal{F} = 1$ characterizes a measurement at the spin-projection noise limit, and $1/\mathcal{F}$ denotes the measurement uncertainty relative to that limit. Alternative readout techniques have been developed to increase measurement fidelity, but most have focused on single spins and small ensembles[7–16], which limits their utility for high-sensitivity measurements[6]. Additionally, these techniques either introduce substantial overhead time[7–13] (diminishing achievable sensitivity) or offer only modest improvements over conventional optical readout[14,16,17].

In this work, we demonstrate a non-optical readout technique for solid-state spin-ensemble sensors. Our technique leverages strong collective coupling between a dielectric resonator cavity and a spin ensemble at room temperature. Similar coupled spin–cavity systems have recently been harnessed to demonstrate a room-temperature maser[18] and Dicke superradiance[19,20]. Related cavity quantum electrodynamics (CQED) effects have also been employed for quantum information applications in cryogenic solid-state[21–28] and superconducting qubit[29–31] systems. Cavities also have been used to great effect in electron paramagnetic resonance (EPR) to amplify weak signals from samples under study[32], including for the observation of the spectrum of a nitrogen-vacancy (NV) center in diamond on illumination with light[33]. Quantitative EPR spectroscopy remains an area of active research for biological, medical, and industrial applications[34]. Here, we report the use of a strongly coupled, room-temperature spin–cavity system for sensor applications, providing in detail new insights into optimization of such systems for sensing. We

demonstrate this technique in a magnetometer using an ensemble of NV⁻ centers in diamond, though the method has broad applicability to any paramagnetic defect with a microwave (MW) resonance (provided there is a means of inducing spin polarization). In addition to providing unity measurement contrast and circumventing the shot-noise limitation inherent to conventional optical spin readout, the readout method introduces no substantial overhead time to measurements and results in an advantageous cavity-mediated narrowing of the magnetic resonance features. Moreover, this advance promises what has long been elusive for quantum sensors based on solid-state spin ensembles: a clear avenue to readout at the spin-projection limit. Because the sensor's limiting noise source is independent of the number of polarized spin defects $N$, the device's sensitivity is expected to improve linearly with increasing $N$ until the spin-projection limit is reached.

The technique, which we term MW cavity readout, operates by measuring changes in an applied MW field following cavity-enhanced interactions with a spin ensemble. When the MW frequency is tuned near-resonant with the spin defect's resonance frequency, both absorptive and dispersive interactions occur[35]. These interactions encode the spin resonance in the amplitude and phase of the transmitted or reflected MWs. While the absorptive and dispersive interactions may be too weak on their own to cause perceptible changes in the MW field, even for a sizeable spin ensemble, these effects can be enhanced more than ten-thousandfold by placing the ensemble in a high-quality-factor cavity resonant with the applied MWs. Dispersion and absorption by the spin ensemble then modify the resonance frequency and linewidth of the composite cavity-spin system, respectively. Consequently, detection of the transmission through or reflection from the composite cavity provides readout of the spin resonance[34].

## Results

**Device Operation.** In the experiments described here, NV⁻ defects are continuously initialized by applying 532 nm laser light. This optical pumping preferentially populates the spin-1 NV⁻ ground-state sublevel $|m_s = 0\rangle$, spin-polarizing the NV⁻ ensemble, as shown in the energy level diagram in Fig. 1a. At zero magnetic field, the defect has a splitting $D \approx 2.87$ GHz between the

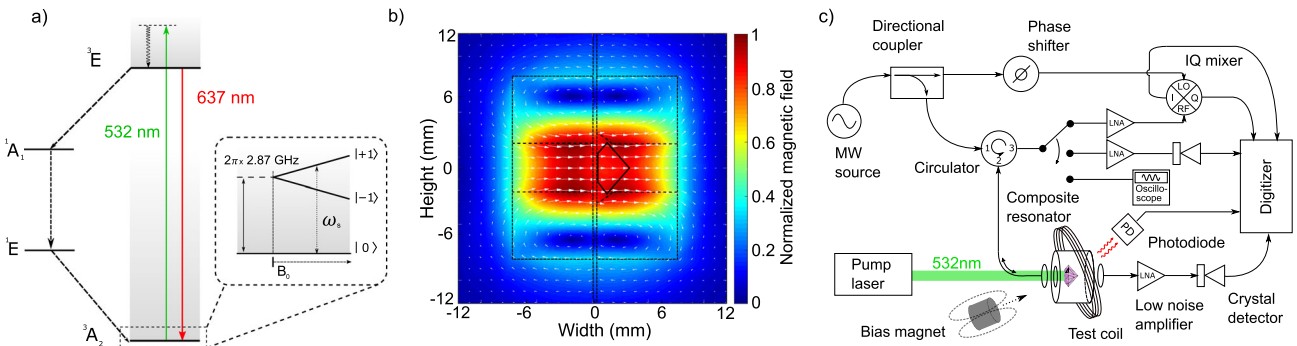

**Fig. 1 Experimental setup for MW cavity readout of NV⁻ centers in diamond. a** Level diagram. The NV⁻ ground-state spin triplet ($^3A_2$) exhibits a 2.87 GHz zero-field splitting between the $|m_s = 0\rangle$ and degenerate $|m_s = \pm1\rangle$ states. This degeneracy may be lifted by application of a bias magnetic field $B_0$, allowing individual addressing of either the $|m_s = 0\rangle \leftrightarrow |m_s = -1\rangle$ or $|m_s = 0\rangle \leftrightarrow |m_s = +1\rangle$ transitions. Optical pumping with 532 nm light initializes spins to the $|m_s = 0\rangle$ state via a non-radiative decay path ($^1A_1 \rightarrow {}^1E$). **b** Microwave cavity magnetic field. Interactions between the interrogation photons and the NV⁻ ensemble can be enhanced by placing the diamond inside a cavity resonant with the applied photons. As illustrated in the axial cut of the composite cavity, the diamond (solid black line) is placed near the antinode of the magnetic field (white arrows) created by the two dielectric resonators (black dashed lines). **c** Device schematic. Applied MWs near-resonant with both the cavity and spin transitions are split into a signal component which interrogates the composite cavity through a circulator (lower branch) and a reference component (upper branch). Microwaves reflected from the composite cavity are amplified before being mixed with the reference component by an IQ mixer whose dual outputs are digitized. Alternatively, reflected MWs can be read out via a MW crystal detector or measured directly using an oscilloscope with a sufficiently high sampling rate. Transmission measurements employ only an amplifier and a crystal detector. A photodiode monitoring red fluorescence allows simultaneous optical readout.

$|m_s = 0\rangle$ state and the $|m_s = \pm 1\rangle$ states. Application of a tunable bias magnetic field $\vec{B}_0$ lifts the degeneracy of the $|m_s = \pm 1\rangle$ states, allowing either of the $|m_s = 0\rangle \leftrightarrow |m_s = \pm 1\rangle$ MW transitions to be individually addressed spectroscopically. The external bias field $\vec{B}_0$ is oriented along the diamond's $\langle 100\rangle$ axis to project equally onto all four NV$^-$ orientations. The MWs are applied with drive frequency $\omega_d$ near-resonant with the $|m_s = 0\rangle \leftrightarrow |m_s = +1\rangle$ transition (with resonance frequency $\omega_s$), and we restrict our discussion to the effective two-level system formed by these states.

The composite MW cavity consists of two concentric cylindrical dielectric resonators surrounding a high-NV$^-$-density diamond mounted on a mechanical support wafer. We define the bare cavity resonance frequency $\omega_c$ as the resonance frequency of the system in the absence of laser-induced spin polarization. Positioning the diamond at the MW magnetic field antinode, as shown in Fig. 1b, maximizes the ensemble-photon coupling. An adjustable input coupling loop couples the MW field into the composite cavity. A circulator allows for reflection measurements, while a supplementary output coupling loop allows for transmission measurements, as depicted in Fig. 1c. The composite MW cavity exhibits an unloaded quality factor of $Q_0 = 22{,}000$.

For magnetometry, the applied MW drive frequency $\omega_d$ is tuned to the bare cavity resonance $\omega_c$. The bias field magnitude $B_0$ is set so that $\omega_s = \omega_c$. Small changes in $B_0$, representing the test magnetic field to be detected, cause $\omega_s$ to vary about $\omega_c$. These changes in $B_0$ (and thus $\omega_s$) are detected by monitoring MWs reflected from the cavity. To understand the readout mechanism, we first consider only the dispersive effect of the NV$^-$ ensemble, neglecting the effect of absorption. (This simplification is valid for sufficiently high-MW power, where the absorptive effect is suppressed relative to the dispersive effect; see Supplementary Note 5.) With $\omega_s = \omega_c$ (and neglecting absorption), reflection from the cavity remains unchanged regardless of the state of the NV$^-$ ensemble (e.g., regardless of whether optical spin-polarization light is applied). As $\omega_s$ shifts away from $\omega_c$, however, the NV$^-$ ensemble produces a dispersive shift that modifies the composite cavity's resonance frequency, resulting in an increase in reflected MW power. Moreover, the dispersive effect produces a phase shift in the reflected voltage $\Gamma V_{\text{In}}$ relative to the incident MWs (where $\Gamma$ is the complex reflection coefficient and $V_{\text{In}}$ is the incident MW voltage), and the sign of this phase shift depends on the sign of $\omega_s - \omega_c$. This allows the use of a phase-sensitive measurement technique by monitoring the quadrature port of an IQ mixer. Because the voltage on this port changes sign for deviations of $\omega_s$ above or below $\omega_c$, with a zero-crossing for $\omega_s = \omega_c$, this measurement technique inherently provides unity contrast (see Methods).

**Spin–cavity interaction**. The interaction between a MW photon and a single spin is described by the Jaynes–Cummings Hamiltonian[36],

$$\mathcal{H} = \hbar\omega_c \hat{a}^\dagger \hat{a} + \frac{1}{2}\hbar\omega_s \hat{\sigma}_z + \hbar g_s\left(\hat{a}^\dagger \hat{\sigma}^- + \hat{a}\hat{\sigma}^+\right), \quad (1)$$

where $\hat{a}^\dagger$ and $\hat{a}$ are the creation and annihilation operators, respectively (for photons at the bare cavity frequency $\omega_c$); $\omega_s$ is the spin resonance frequency; and $\hat{\sigma}_z$, $\hat{\sigma}^+$, and $\hat{\sigma}^-$ are the Pauli-z, raising, and lowering operators. The single-spin–photon coupling $g_s$ at the cavity antinode is[37–39] $g_s = \frac{\gamma}{2}\mathfrak{n}_\perp\sqrt{\frac{\hbar\omega_c \mu_0}{V_{\text{cav}}}}$, where $\gamma$ is the electron gyromagnetic ratio, $\mu_0$ is the vacuum permeability, $\hbar$ is the reduced Planck constant, and $V_{\text{cav}}$ is the mode volume of the microwave cavity resonance. The coefficient $\mathfrak{n}_\perp \leq 1$ is a geometrical factor, which is required because only the component of the cavity field transverse to the spin quantization axis can drive transitions (and the spin quantization axis may be set by a

crystallographic axis, at an energy scale much greater than that of the coupling between the magnetic field and the spin). When the cavity and spin resonances are nearly degenerate, which is the regime employed in this work, the hybridized spin–cavity modes result in the familiar spectroscopic feature known as Rabi splitting.

For an ensemble of $N$ polarized spins, the Jaynes–Cummings model is generalized to the Tavis–Cummings model[40,41], with $g_s$ replaced by the effective collective coupling $g_{\text{eff}} = g_s\sqrt{N}$[42]. Predictions of this model are consistent with measurements of the MW response of solid-state spin ensembles strongly coupled to dielectric resonators at room temperature[18,19]. Since the MW cavity magnetic field varies by only a small amount ($\approx\pm 3.5\%$) over the diamond volume, we assume each spin has an identical coupling strength $g_s$.

In order to provide a connection with the physical parameters of the experimental apparatus, it is convenient to develop a description of the system in terms of an equivalent circuit model. (The derivation of which is described in Supplementary Note 3.) The resulting RLC circuit model provides expressions for the reflection and transmission coefficients, which can then be formulated in terms of the quantum mechanical parameters of the system. With an ensemble undergoing constant optical-pumping-induced spin polarization at a rate $\kappa_{\text{op}} = 1/T_1^{\text{op}}$, the voltage reflection coefficient is given by

$$\Gamma = -1 + \frac{\kappa_{c1}}{\frac{\kappa_c}{2} + j(\omega_d - \omega_c) + \frac{g_{\text{eff}}^2}{\frac{\kappa_s}{2} + j(\omega_d - \omega_s) + \frac{g_s^2 n_{\text{cav}}\cdot\kappa_s/(2\kappa_{\text{op}})}{\frac{\kappa_s}{2} - j(\omega_d - \omega_s)}}}, \quad (2)$$

where the cavity loss rate $\kappa_c \equiv \kappa_{c0} + \kappa_{c1} + \kappa_{c2}$ is the sum of the unloaded, input port, and output port loss rates, respectively; $\kappa_s = 2/T_2$ is the homogeneous width of the spin resonance (with decoherence time $T_2$); and $n_{\text{cav}}$ is the average number of cavity photons. (See Methods for the corresponding expression for the transmission coefficient and the Supplementary Note 3 for additional information on the derivation of these expressions.) Here, to simplify the presentation, we have omitted in (2) integration over the inhomogeneous distribution of spin resonance frequencies; this distribution can be included following the methods of refs. [43,44]. We find that the inhomogeneous linewidth must be accounted for to produce optimal agreement in numerical models used to fit the experimental data.

Neglecting absorption, the imaginary part of the reflection coefficient can be approximately expressed in a more illuminating form within a particular regime relevant to magnetometry. For critical input coupling ($\kappa_{c1} = \kappa_{c0}$), no output coupling ($\kappa_{c2} = 0$), and $\omega_d = \omega_c$, the reflection coefficient in the limiting case of small spin–cavity detunings ($|\omega_s - \omega_c| \ll \kappa_s/2$) is approximately given by

$$\text{Im}\left[\Gamma\right] \approx \frac{8g_{\text{eff}}^2}{(\kappa_s^*)^2\kappa_c}(\omega_c - \omega_s), \quad (3)$$

where $\kappa_s^*$ characterizes the inhomogeneous linewidth. This approximate expression is valid for $n_{\text{cav}}$ high enough to saturate the homogeneous linewidth $n_{\text{cav}} \gg \frac{\kappa_{op}\kappa_s}{2g_s^2}$ but below the number to produce substantial power broadening $n_{\text{cav}} \lesssim \frac{\kappa_{op}\kappa_s^*}{2g_s^2}$. Equation (3) highlights the potential of this technique for high-sensitivity magnetometry, as $\text{Im}[\Gamma]$ is proportional to spin–cavity detuning.

The prefactor $\frac{8g_{\text{eff}}^2}{(\kappa_s^*)^2\kappa_c}$ in (3) is closely related to the collective cooperativity, a dimensionless figure of merit for the ensemble-cavity coupling strength typically defined as $\xi = \frac{4g_{\text{eff}}^2}{\kappa_s\kappa_c}$[45]. To maximize spin readout fidelity, it is important to engineer the cooperativity of the ensemble-cavity system to be as large as possible. The system's cooperativity is experimentally determined

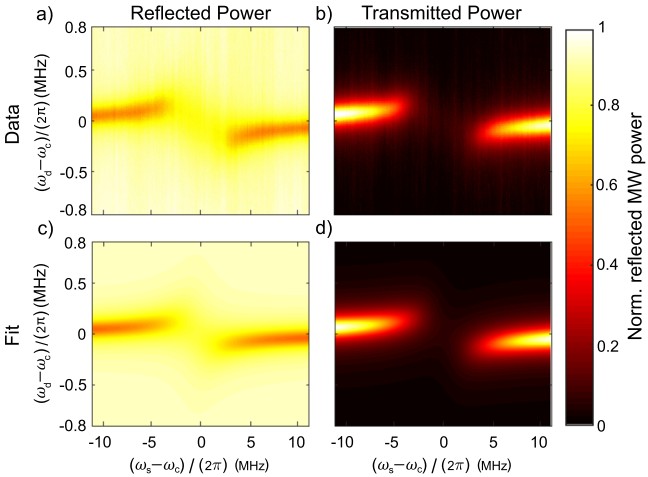

**Fig. 2 Strong ensemble-cavity coupling under ambient conditions.** The spin resonance frequency is swept relative to the bare cavity resonance (horizontal axis) by varying the applied magnetic field; simultaneously varying the MW drive frequency (vertical axis) reveals the spin-ensemble-modified composite cavity resonance. Data are recorded both in reflection (**a**) and transmission (**b**). The data are fit (**c**, **d**) to (2) and (4) using a 2D nonlinear least-squares solver. The fit gives $g_{eff} = 2\pi \times 0.70$ MHz; see Methods for additional fit parameters. Each plot is normalized to unity, and recorded data are taken with $-56$ dBm of MW drive power.

from the avoided crossing observed in recorded reflected and transmitted MW power, which are measured as the spin resonance frequency $\omega_s$ and MW drive frequency $\omega_d$ vary with respect to the bare cavity resonance $\omega_c$. These measurements, shown in Fig. 2, are performed at low MW drive power to avoid perturbing the system. For the data in Fig. 2, both coupling loops are under-coupled, resulting in a full-width-half-maximum (FWHM) loaded cavity linewidth of $\kappa_c = 2\pi \times 200$ kHz (given the measured loaded quality factor $Q_L = 14{,}500$). We extract $2g_{eff} = 2\pi \times 1.4$ MHz (see Methods). Because the spin resonance linewidth arises from both homogeneous (e.g., dipolar interactions) and inhomogeneous (e.g., strain) mechanisms, with differing effects on the behavior of the system (see Supplementary Notes 3 and 5), the appropriate value of $\kappa_s$ for calculating the cooperativity is not obvious. We model the cooperativity, including inhomogeneous broadening, using the method of ref. [46] (see Methods). This analysis produces a value $\xi = 1.8$ under the experimental conditions used for measurement (i.e., $\kappa_c = 2\pi \times 200$ kHz) or $\xi = 2.8$ assuming negligible losses to input and output coupling (i.e., $\kappa_c = \kappa_{c0}$).

**Cavity-enhanced magnetometry.** While useful for characterizing spin–cavity coupling strength, operation at low applied MW power is undesirable for high-fidelity spin readout due to the fixed contribution of Johnson noise. Applying higher MW power minimizes the fractional contribution of Johnson noise and other additive noise sources, but higher applied power will also produce deleterious broadening of the spin ensemble resonance; the optimum power is set by a balance between these two considerations (see Methods). We empirically determine that approximately 10 dBm is optimal for the present system (see Supplementary Note 5), resulting in a maximum reflected power of $-2.4$ dBm. The high peak reflected MW power ($3.0 \times 10^{20}$ MW photons/s) for the NV$^-$ ensemble of $\approx 1.4 \times 10^{15}$ polarized spins, combined with unity contrast, ensures that MW photon shot noise does not limit the achievable readout fidelity (given experimentally relevant readout timescales; see Supplementary Note 1).

The readout method also provides a cavity-mediated narrowing of the magnetic resonance feature. This narrowing is illustrated in

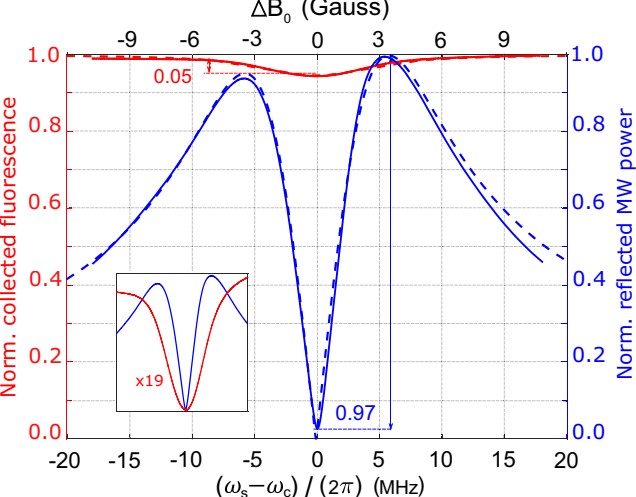

**Fig. 3 Comparison of contrast and linewidth in MW cavity readout magnetic resonance and ODMR.** The signal associated with the NV$^-$ $|m_s = 0\rangle \leftrightarrow |m_s = +1\rangle$ magnetic resonance is recorded simultaneously using MW cavity readout (blue solid line) and conventional optical readout (red solid line). The MW cavity readout realizes contrast $C = 0.97$, limited by imperfect circulator isolation, while conventional optical readout realizes contrast $C = 0.05$ (see Methods). For ease of comparison with the ODMR lineshape, MW cavity readout is performed here using a phase-insensitive measurement of reflected MW power, rather than the phase-sensitive technique; see Methods. Fits from the inhomogeneously-broadened numerical model (blue dashed line) and a Lorentzian model of ODMR (red dashed line) are also shown; see Supplementary Note 5. All $^{14}$N hyperfine transitions are included in both models, but the hyperfine structure is not resolved due to the substantial inhomogeneous broadening. The inset shows both readout signals scaled to the same peak-to-peak values, highlighting the $\approx 2 \times$ narrowing of the magnetic resonance feature observed with MW cavity readout. The left-right asymmetry in the MW cavity readout signal is attributed to $\approx -20$ kHz detuning of the applied microwaves from the bare cavity resonance. The applied MW power is 10 dBm.

Fig. 3, which shows a MW cavity readout magnetic resonance signal plotted alongside a conventional optically detected magnetic resonance (ODMR) signal recorded simultaneously. The MW cavity readout feature exhibits a FWHM linewidth of 4 MHz, while the ODMR linewidth is 8.5 MHz (FWHM). To understand this narrowing, consider the resonance feature associated with reflection from the bare cavity (i.e., the composite cavity without laser light applied) vs. MW drive frequency $\omega_d$. The cavity linewidth $\kappa_c$ is independent of the spin resonance linewidth $\kappa_s$ and, in principle, can be made narrower than the spin resonance by improving the cavity quality factor $Q_0$. The linewidth of the cavity-mediated magnetic resonance feature, however, is a function of both the cavity linewidth and the spin resonance linewidth; roughly speaking, the former determines the dispersive shift needed to reflect 80% input power, while the latter partially determines the size of the dispersive shift for a given change in magnetic field. Moreover, the size of the dispersive shift for a given change in magnetic field is not determined solely by the spin resonance linewidth; the size of this shift increases with increased cooperativity. Thus, the cavity-mediated linewidth can be narrower than the spin resonance linewidth for sufficiently large values of $g_{eff}$ and sufficiently small values of $\kappa_c$. The cavity-mediated narrowing is advantageous to magnetometer operation, as narrower magnetic resonance features can be localized with greater precision. The line narrowing effect is in agreement with

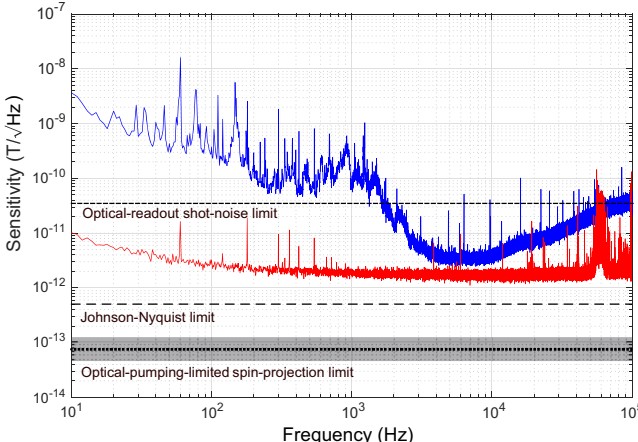

**Fig. 4 MW cavity readout magnetometer sensitivity.** Based on noise spectral density measured during magnetometer operation (blue solid line), we project a sensitivity of $\approx 3$ pT/$\sqrt{\text{Hz}}$ in the band from 5 kHz to 10 kHz, where sensitivity approaches the limit set by the measured noise floor of the amplifier and digitizer electronics (red solid line). Also depicted are the optical-readout shot-noise limit (black short dashed line) of the experimental setup, the calculated Johnson-Nyquist noise limit (black long dashed line) of 0.5 pT/$\sqrt{\text{Hz}}$ and the optical-pumping-limited spin-projection limit (black dotted line). The optical-pumping-limited spin-projection limit is bounded above and below (gray shaded box) to illustrate uncertainty arising from estimating the optical pumping relaxation time $T_1^{\text{op}}$ (see Methods). Magnetometry is performed using the phase-sensitive technique of recording reflected MW voltage through the IQ mixer; IQ traces are shown in Supplementary Note 5.

expected behavior from the numerical model including inhomogeneous broadening, as shown in Fig. 3.

The magnetometer is calibrated with a 10 Hz test magnetic field with a 1 $\mu$T root-mean-square (RMS) amplitude. The measured noise spectrum is scaled using this known magnetic field value to produce a noise spectrum in magnetic field units, and we project a minimum sensitivity of 3.2 pT/$\sqrt{\text{Hz}}$ from approximately 5 to 10 kHz (see Fig. 4). For a 1 nT test field measured over 1 s, this projected minimum sensitivity corresponds to a signal-to-noise ratio (SNR) of $\approx 310$, compared to a photon-shot-noise-limited SNR of $\approx 40$ for optical readout in this apparatus. In future work, DC signals can be upmodulated to this low-noise band by the application of an AC magnetic bias field.

The projected sensitivity of the present magnetometer is among the highest reported broadband sensitivities of devices employing NV ensembles. For example, the best NV-ensemble-based broadband magnetometers employing conventional optical readout have achieved sensitivities ranging from 0.9 pT/$\sqrt{\text{Hz}}$[47] to 15 pT/$\sqrt{\text{Hz}}$[48]. The projected sensitivity using MW cavity readout is limited by the phase noise of the interrogation microwaves, Johnson-Nyquist (thermal) noise, and vibration-induced changes in the coupling to the composite cavity. Phase noise manifests as frequency fluctuations, which cause variations in reflected power unrelated to the magnetic field value. Selection of a lower-phase-noise MW source would reduce these fluctuations. Vibration-induced fluctuations could be reduced by engineering a more robust cavity coupling mechanism. Together, these changes could allow the Johnson-Nyquist-noise limit of 0.5 pT/$\sqrt{\text{Hz}}$ (see Fig. 4) to be reached. Crucially, unlike shot noise, these limiting noise sources remain fixed as the signal strength increases. Therefore, there exists a straightforward path to improving sensitivity toward the spin-projection limit: augmenting the signal through increasing the collective

cooperativity $\xi$. Cooperativity can be improved by increasing the number of polarized spins, increasing the cavity quality factor, or reducing the spin-resonance linewidth[49] (see Methods). Furthermore, pulsed measurement protocols could be employed to avoid sensitivity degradation due to MW power broadening. We note that, shortly after the preprint of this work was reported, complementary work employing related pulsed techniques was reported independently in ref. [50].

## Discussion

The MW cavity readout method demonstrated here offers compelling advantages over alternative approaches for bulk solid-state quantum sensors. First, the method realizes unity contrast and circumvents the photon shot-noise limitations inherent to conventional optical readout. In addition, unlike alternative optical readout techniques, MW cavity readout does not introduce deleterious overhead time in the measurement process. Finally, the technique promises favorable scaling; the measurement SNR increases linearly with the number ($N$) of defects interrogated, allowing for readout at the spin-projection limit for sufficiently large $N$. Room-temperature magnetometry with sensitivity approaching the spin-projection limit would enable an increase in the utility of solid-state quantum sensors, for example in magnetocardiography[51] and magnetoencephalography[52] devices. Although demonstrated here using NV$^-$ centers in diamond, MW cavity readout can be performed on other solid-state crystals and paramagnetic spins, and is not exclusive to the small minority demonstrating optical fluorescence with significant spin-state dependence. For example, divacancy[53] and silicon-vacancy centers[54] in silicon carbide can be optically spin polarized, but these defects display poor fluorescence contrast between spin states[55]; thus, cavity-enhanced MW readout could offer advantages for sensors based on these defects. In addition to magnetometry, we expect that this technique will find broad application in precision tests of fundamental physics[56], precision frequency generation[18], and electric field sensing[2,57].

## Methods

**Experimental setup**. This work employs a natural, brilliant-cut diamond with volume $V_{\text{dia}} = 25$ mm$^3$ which was subsequently HPHT-processed and irradiated following the Lucent process[58]. From electron paramagnetic resonance (EPR) measurements and comparison with a reference sample, the NV$^-$ density is estimated to be [NV$^-$] = 5 ± 2.5 ppm, corresponding to a total NV$^-$ number $N_{\text{tot}} = 2 \pm 1 \times 10^{16}$. As a natural diamond, the sample displays substantial strain and exhibits an inhomogeneous dephasing time $T_2^*$ of 40 ns. The P1 centers (as interrogated via EPR) exhibit a full-width-half-maximum linewidth of 910 kHz, of which approximately 300 kHz can be attributed to broadening from $^{13}$C spins[6]. The residual 610 kHz linewidth suggests an approximate total nitrogen concentration [N$^T$]=18 ppm[59], while integration of the P1 EPR signal suggests [N$_s^0$] = 22 ppm. For simplicity we assume [N$^T$] = 20 ppm, which corresponds to an estimated NV$^-$ decoherence time $T_2 = 8$ $\mu$s. The value of [NV$^0$] is evaluated using the method of Alsid et al. to be [NV$^0$] = 1 ± 0.5 ppm[60].

The diamond is affixed to a semi-insulating wafer of silicon carbide (SiC) for mechanical support and located coaxially between two cylindrical dielectric resonators (relative dielectric $\epsilon_r \approx 34$, radius $a = 8.17$ mm, cylindrical length $L = 7.26$ mm, with a 4 mm diameter center-cut hole). The combined diamond-resonator composite cavity has a resonance frequency $\omega_c = 2\pi \times 2.901$ GHz and an unloaded quality factor $Q_0 \approx 22000$. The composite cavity is centered inside an aluminum shield (inner diameter = 50.8 mm, length = 89 mm) to reduce radiative losses. NV$^-$ centers within the diamond are continuously polarized into the $|m_s = 0\rangle$ Zeeman sublevel energy level by approximately 12 W of 532 nm optical excitation. A neodymium-iron-boron permanent magnet applies a 19.2 G static magnetic field $\vec{B}_{\text{perm}}$ along the diamond's $\langle 100 \rangle$ axis. An additional test coil applies a tunable magnetic field ($\vec{B}_{\text{coil}}$) along the same direction; the total bias field $\vec{B}_0$ can then be varied over the range 19.2 ± 25 G.

Figure 1c depicts the main MW circuit components. Microwaves (produced by a Keysight E8257D PSG) at frequency $\omega_d$ are split into a signal and reference component, with the signal components passing through an attenuator and circulator before coupling into the composite cavity. The MWs are inductively coupled to the composite cavity by a wire loop (the input coupling loop) mounted on a translation stage. MWs reflected from the cavity can be measured in one of three ways: directly via the 50 $\Omega$ termination of an oscilloscope; through an

amplifier followed by a crystal detector (which measures a correlate of the reflected power); or through an amplifier to the RF port of an IQ mixer, with the local oscillator (LO) port driven by the reference MW component. Transmission occurs through an additional wire loop (the output coupling loop) on a translation stage and is measured on a crystal detector.

Slight modifications of the setup are employed to collect the data shown in Figs. 2, 3 and 4, as described below.

**Strong coupling.** Reflection and transmission data in Fig. 2 are collected simultaneously. For both transmission and reflection measurements, the MWs are detected using a crystal detector operating in the linear regime. During this measurement, both the input and output coupling loops are undercoupled ($Q_L = 14,500$, compared to $Q_0 = 22,000$). $\overrightarrow{B}_{coil}$ is increased from approximately $-6.8$ G to $+6.8$ G (altering $\omega_s$) in steps of 0.068 G while the MW drive $\omega_d/(2\pi)$ is varied relative to $\omega_c/(2\pi)$ over the range $-800$ kHz to $+800$ kHz. At each step of the bias field ($\overrightarrow{B}_{coil}$) sweep and at each MW drive frequency, the reflected and transmitted MWs are measured. The 2D power data are then fit to the square of the voltage reflection (equation (2)) and the square of the voltage transmission, given by

$$ T = \frac{\sqrt{\kappa_{c1}\kappa_{c2}}}{\frac{\kappa_s}{2} + j(\omega_d - \omega_c) + \frac{g_{eff}^2}{\frac{\kappa_s}{2} + j(\omega_d - \omega_s) + \frac{g_s^2 n_{cav} \cdot \kappa_s/(2\kappa_{op})}{\frac{\kappa_s}{2} - j(\omega_d - \omega_s)}}}. \qquad (4) $$

The reflection and transmission coefficients are consistent with those derived from a quantum mechanical treatment of the electromagnetic field[61–63] using input-output theory[64,65]. The final fit parameters are $g_{eff} = 2\pi \times 0.70$ MHz, $\kappa_{c0} = 2\pi \times 125$ kHz, $\kappa_{c1} = 2\pi \times 25.3$ kHz, $\kappa_{c2} = 2\pi \times 33.4$ kHz, and $\kappa_s = 2\pi \times 5.24$ MHz. Here, the fit $\kappa_s$ should be interpreted as an effective linewidth including inhomogeneous broadening; see Supplementary Notes 3 and 5.

**Cavity-mediated narrowing and contrast.** The data in Fig. 3 are also collected employing the crystal detector to measure reflected MW power. The MW drive is set to the bare cavity resonance, $\omega_d = \omega_c$. The input coupling loop is critically coupled to the composite cavity, and the output coupling loop is removed, so that $\kappa_c = 2\kappa_{c0}$. The spin transition frequency $\omega_s$ is tuned across the cavity resonance $\omega_c$ by varying the value of $\overrightarrow{B}_{coil}$ as detailed above. An auxiliary photodiode allows simultaneous measurement of the NV$^-$ fluorescence signal. In this measurement configuration, the contrast is slightly below unity due primarily to the imperfect isolation of the MW circulator. (For CW measurements, as performed here, we define the contrast $C = \frac{a-b}{a}$ where $a$ and $b$ denote the respective maxima and minima signal values when the bias field is swept over the magnetic resonance.)

**Magnetometry measurements and sensitivity.** For magnetometry, MWs reflected from the composite cavity are amplified, band-pass filtered, and mixed with an attenuated and phase-shifted reference component. The reflected signal is mixed to base band using an IQ mixer. The phase of the reference component, which drives the mixer local oscillator (LO) port, is adjusted until the absorptive ($\propto \mathrm{Re}[\Gamma]V_{In}$) and dispersive ($\propto \mathrm{Im}[\Gamma]V_{In}$) components are isolated to the in-phase (I) and quadrature (Q) channels respectively.

The magnetometry sensisitivity is characterized by monitoring the Q channel as a 1 $\mu$T (RMS) field is applied via the test coil. The test field is calibrated using the known dependence of the ODMR resonances on the applied field. The RMS amplitude of the test field is checked with a commercial magnetometer and also via calculation from the known coil geometry and applied current. The magnetometer sensitivity is given by

$$ \eta = \frac{e_n}{V_{Dig}/B_{test}^{RMS}}, \qquad (5) $$

where $e_n$ is the RMS voltage noise floor (at the digitizer) of the double-sided spectrum (20 nV/$\sqrt{\mathrm{Hz}}$, which occurs between 5 and 10 kHz), $B_{test}^{RMS}$ is a 1 $\mu$T RMS amplitude magnetic field at 10 Hz frequency, and $V_{Dig}$ is the RMS voltage recorded at the digitizer in response to the test magnetic field.

Although applying higher MW power decreases fractional Johnson noise, it also broadens the dispersive resonance feature[66]. Hence, there exists an optimal power $P$ to achieve a maximum absolute value of the slope $|d(\mathrm{Im}[\Gamma]V_{RMS})/d\omega_s|$ (where $V_{RMS}$ is the RMS incident MW voltage) and thus maximal sensitivity to changes in $\omega_s$. For the present system, we empirically determine that $P = 10$ dBm is optimal (see Supplementary Note 5), which results in a maximum reflected power of $-2.4$ dBm.

In the high-MW-drive-power (i.e., primarily dispersive) regime, the maximal slope is achieved in the Q channel when $\omega_s = \omega_c = \omega_d$. By using only the permanent magnet to set $\omega_s = \omega_c$, we ensure that the test coil current source does not contribute to the noise floor of the magnetometer.

## Data availability
The data in Figs. 1–4 that support the findings of this study are available from the corresponding author upon reasonable request.

## Code availability
The code that supports the findings of this study are available from the corresponding author upon reasonable request.

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

## Acknowledgements

The authors acknowledge L. M. Pham and J. A. Majumder for helpful discussions and assistance in determining properties of the experimental sample, R. McConnell for useful discussions on circuit and cavity quantum electrodynamics, and C. Panuski for helpful discussions on EPR and early theory developments. E.R.E. was supported by the National Science Foundation (NSF) through the NSF Graduate Research Fellowships Program. This material is based upon work supported by the Under Secretary of Defense for Research and Engineering under Air Force Contract No. FA8702-15-D-0001. Any opinions, findings, conclusions or recommendations expressed in this material are those of the author(s) and do not necessarily reflect the views of the Under Secretary of Defense for Research and Engineering.

## Author contributions

J.F.B. and M.F.O. conceived the project. E.R.E. and J.F.B. designed and constructed the experimental apparatus, with D.A.B. providing technical guidance. E.R.E. performed the experiments and developed analysis software. M.F.O. developed the theory, with J.F.B., E.R.E., J.M.S., and D.A.B. providing additional theory development. E.R.E., J.M.S., M.H.S., J.F.B., and M.F.O. prepared the manuscript and contributed to data analysis. All authors discussed results and revised the manuscript. D.A.B. and D.R.E. supervised the project.

## Competing interests

The authors declare no competing interests.
