## [Peer Review File · Nature Communications]

REVIEWER COMMENTS

Reviewer #1 (Remarks to the Author):

Eisenach et al. report an interesting demonstration of the readout of an ensemble of NV centers in diamond through microwaves, by coupling the system to a microwave cavity. This is technically impressive, and conceptually interesting, work and is step forward to some exciting applications, such as improved magnetometry with room temperature spin ensembles. I therefore recommend the manuscript for publication in *Nature Communications*.

Reviewer #2 (Remarks to the Author):

Eisenach and coworkers present a room-temperature microwave detection of an ensemble of nitrogen-vacancy (NV-) centers via its strong coupling to a dielectric microwave cavity. They subsequently evaluate this readout technique for magnetometry and estimate a magnetic sensitivity of $3 \text{ pT}/\sqrt{\text{Hz}}$ over the 10 kHz - 50 kHz range and $\sim 1 \text{ nT}/\sqrt{\text{Hz}}$ for DC magnetic fields. This approach is rather interesting for the field of NV center magnetometers and deserved to be underlined, since one of the factors limiting their sensitivity is the poor collection of the spin ensemble fluorescence signal using the NV center conventional optical readout.

Compared to the first version of this manuscript, Eisenach and coworkers have clarified the motivation and context of their work, and the changes have been beneficial. However, I (still) have the following remarks and questions:

- Why did the authors choose contrast for comparison between microwave and optical readout? As the acquisition protocol is rather different, I expect both measurements to have different SNR, and thus a simple comparison of their contrasts does not seem to be the most relevant to evaluate the sensitivity of each readout technique. I thus find the introduction and analysis of Figure 3 rather confusing. Could the authors at the very least precise the SNR of each measurement and compare the sensitivity of each technique for their setup ?

- Authors use 'high-fidelity readout' expression: this is rather well defined for a single quantum object, but what does it mean for an ensemble detection?

- Could the authors compare explicitly their sensitivity results/spin readout projective with other magnetometers based on NV centers ensemble, especially since their optical readout seems rather under-coupled compared to the state-of-the-art?

- Fig 4: where would the photon-shot noise limit in case of optical readout be located on this graph ? below or above the Johnson-Nyquist noise limit ?

- There is a confusion existing in the manuscript between the dispersive and absorptive components of the signal reflecting on the spins+cavity system, and the resonant vs dispersive regime of CQED. Since the work of the authors take place in the regime of resonant CQED, it would be clearer to remove any reference to the dispersive regime of CQED. In particular, in the 2nd paragraph of part 2B, the authors refer to the Rabi splitting and dispersive shift occurring a single spin strongly coupled to a cavity, which is not the regime in which this work takes place. I recommend the authors simply remove the first sentences of this paragraph.

Reviewer #3 (Remarks to the Author):

In their revised manuscript, Eisenach et al. have responded to our comments thoroughly. In particular, they addressed the major issue regarding the experiments' interpretation. The revised discussion and modeling, including both classical and quantum-mechanical interpretations, accurately represent the experiment. Due to the impressive absolute magnetic sensitivity and the thoroughness of the analysis, the manuscript should be well received by the diverse readership of Nature Communications.

We noted minor points that should be addressed prior to publication:

1. P3 “At large spin-cavity detunings, the coupling produces a dispersive shift without significant spontaneous emission, which can be harnessed for quantum non-demolition state readout [29, 40] enables high-fidelity readout of superconducting qubits [30].”

It seems that the word “and” is missing before “enables high-fidelity readout...”

2. P5 “Selection of a lower-phase-noise MW source would reduce these fluctuations.”

The authors should list the model number of their microwave source. Are better sources easily available, or is this a major technical limitation?

We thank the reviewers for their careful reading of and constructive feedback on our manuscript, which has benefited greatly from their attention. Detailed responses to each reviewer's comments are given below. Changes to the text have been color coded in green for clarity.

In addition, we have updated the manuscript to reflect theory developments since our initial submission. These new insights primarily affect the manuscript's supplemental material section. The results in the manuscript remain unchanged. Changes to the manuscript have been color coded in orange for clarity and are briefly outlined here:

- New theory developments have shown that the parallel RLC equivalent circuit (as opposed to the series RLC equivalent) better represents the phase far from resonance of the loop-coupled dielectric cavity. The following equations have been amended to reflect this change: (2), (4), and (37) – (46).
- We have changed the manuscript to reflect the electrical engineering convention of using j to describe the imaginary unit $\sqrt{-1}$. The equations affected are (2), (4), (14) – (17), (19), and (24).
- We have added the full equations for the reflection and transmission coefficients including integration over the inhomogeneous spin resonance distributions. In the updated manuscript, the reflection and transmission coefficient equations are (47) and (48), respectively.

REVIEWER COMMENTS

Reviewer #1 (Remarks to the Author):

Eisenach et al. report an interesting demonstration of the readout of an ensemble of NV centers in diamond through microwaves, by coupling the system to a microwave cavity. This is technically impressive, and conceptually interesting, work and is step forward to some exciting applications, such as improved magnetometry with room temperature spin ensembles. I therefore recommend the manuscript for publication in Nature Communications.

Reviewer #2 (Remarks to the Author):

Eisenach and coworkers present a room-temperature microwave detection of an ensemble of nitrogen-vacancy (NV-) centers via its strong coupling to a dielectric microwave cavity. They subsequently evaluate this readout technique for magnetometry and estimate a magnetic sensitivity of $3 \text{ pT}/\sqrt{\text{Hz}}$ over the $10 \text{ kHz} - 50 \text{ kHz}$ range and $\sim 1 \text{ nT}/\sqrt{\text{Hz}}$ for DC magnetic fields. This approach is rather interesting for the field of NV center magnetometers and deserved to be underlined, since one of the factors limiting their sensitivity is the poor collection of the spin ensemble fluorescence signal using the NV center conventional optical readout.

Compared to the first version of this manuscript, Eisenach and coworkers have clarified the motivation and context of their work, and the changes have been beneficial. However, I (still) have

the following remarks and questions:

- Why did the authors choose contrast for comparison between microwave and optical readout? As the acquisition protocol is rather different, I expect both measurements to have different SNR, and thus a simple comparison of their contrasts does not seem to be the most relevant to evaluate the sensitivity of each readout technique. I thus find the introduction and analysis of Figure 3 rather confusing. Could the authors at the very least precise the SNR of each measurement and compare the sensitivity of each technique for their setup?

We believe that contrast provides a more meaningful and generalizable comparison between the readout techniques, as opposed to SNR, since the SNR depends on both the size of the signal B-field applied and (for optical readout) the efficiency of photon collection. However, given that neither metric provides a full comparison of the readout techniques, we now display the photon-shot-noise-limited sensitivity in Fig. 4, and we now explicitly calculate the SNR values for our technique and conventional optical readout in this apparatus (see the third paragraph of the “Cavity-enhanced magnetometry” section).

- Authors use ‘high-fidelity readout’ expression: this is rather well defined for a single quantum object, but what does it mean for an ensemble detection?

As noted in the introduction to this work (paragraph 1), the readout fidelity in an ensemble is defined as the ratio of the spin-projection limit to the achieved sensitivity. This usage of the term is standard in the NV magnetometry community (see [1] -- [3]).

- Could the authors compare explicitly their sensitivity results/spin readout projective with other magnetometers based on NV centers ensemble, especially since their optical readout seems rather under-coupled compared to the state-of-the-art?

We have expanded our discussion on page 5 to explicitly compare our sensitivity results to the best broadband NV ensemble magnetometers.

- Fig 4: where would the photon-shot noise limit in case of optical readout be located on this graph ? below or above the Johnson-Nyquist noise limit ?

The photon-shot-noise limit for optical readout lies a factor of 20 above the Johnson-noise limit in our experimental apparatus. In Fig. 4, we have added a dashed line to indicate the photon-shot-noise limit realized in our experiments.

- There is a confusion existing in the manuscript between the dispersive and absorptive components

of the signal reflecting on the spins+cavity system, and the resonant vs dispersive regime of CQED. Since the work of the authors take place in the regime of resonant CQED, it would be clearer to remove any reference to the dispersive regime of CQED. In particular, in the 2nd paragraph of part 2B, the authors refer to the Rabi splitting and dispersive shift occurring a single spin strongly coupled to a cavity, which is not the regime in which this work takes place. I recommend the authors simply remove the first sentences of this paragraph.

We have amended the text in part 2B, according to this reviewer's comment, to make it clearer that we are not operating in the dispersive regime of CQED.

Reviewer #3 (Remarks to the Author):

In their revised manuscript, Eisenach et al. have responded to our comments thoroughly. In particular, they addressed the major issue regarding the experiments' interpretation. The revised discussion and modeling, including both classical and quantum-mechanical interpretations, accurately represent the experiment. Due to the impressive absolute magnetic sensitivity and the thoroughness of the analysis, the manuscript should be well received by the diverse readership of Nature Communications.

We noted minor points that should be addressed prior to publication:

1. P3 "At large spin-cavity detunings, the coupling produces a dispersive shift without significant spontaneous emission, which can be harnessed for quantum non-demolition state readout [29, 40] enables high-fidelity readout of superconducting qubits [30]."

It seems that the word "and" is missing before "enables high-fidelity readout..."

We have corrected this sentence.

2. P5 "Selection of a lower-phase-noise MW source would reduce these fluctuations."

The authors should list the model number of their microwave source. Are better sources easily available, or is this a major technical limitation?

We have amended the Methods section to include the model of the microwave source.

In our experiment, we used an E8257D PSG microwave source. Better phase-noise sources are available both as flexible signal generators and as fixed frequency oscillators. However, given the wide array of techniques and continued technological development in microwave signal generation, we do not believe it is appropriate to add further discussion of potential future microwave sources in this work.

References

- [1] *Sensitivity optimization for NV-diamond magnetometry*. John F. Barry, Jennifer M. Schloss, Erik Bauch, Matthew J. Turner, Connor A. Hart, Linh M. Pham, and Ronald L. Walsworth. 2020 REVIEWS OF MODERN PHYSICS. This appears in the manuscript as reference 6.
- [2] *High-sensitivity diamond magnetometer with nanoscale resolution*. J. M. Taylor, P. Cappellaro, L. Childress, L. Jiang, D. Budker, P. R. Hemmer, A. Jacoby, R. Walsworth, M. D. Lukin. 2008 Nature Physics. This appears in the manuscript as reference 1.
- [3] *Nuclear magnetic resonance detection and spectroscopy of single proteins using quantum logic*. I. Lovchinsky, A. O. Sushkov, E. Urbach, N. P. de Leon, S. Choi, K. De Greve, R. Evans, R. Gertner, E. Bersin, C. Müller, L. McGuinness, F. Jelezko, R. L. Walsworth, H. Park, M. D. Lukin. 2016 Science